# Key Considerations to the Introduction of Intergenerational Practice to Australian Policy

**DOI:** 10.3390/ijerph191811254

**Published:** 2022-09-07

**Authors:** Katrina Radford, Janna Anneke Fitzgerald, Nerina Vecchio, Jennifer Cartmel, Ryan Bruce Gould, Jennifer Kosiol

**Affiliations:** 1Department of Employment Relations and Human Resource Management, Griffith University, Nathan 4111, Australia; 2Department of Business, Strategy and Innovation, Griffith University, Gold Coast 4222, Australia; 3Department of Economics, Finance and Accounting, Griffith University, Gold Coast 4222, Australia; 4Department of Social Work and Human Services, Griffith University, Logan 4131, Australia; 5School of Applied Psychology—Health Service Management, Griffith University, Brisbane 4101, Australia

**Keywords:** social policy, intergenerational programs, age friendly communities

## Abstract

Intergenerational practice programs provide purposeful interactions between generations. While research reports improved social and behavioral outcomes for cohorts, no study has explored both expert and potential consumer perceptions of the implementation of intergenerational practice programs. This study conducted a Delphi study of expert opinions, as well as a national survey of potential consumers (N = 1020), to provide critical insights into the potential barriers to implementing intergenerational practice programs. Results revealed that 71.3% of potential consumers would participate in intergenerational practice programs if they were available and experts agreed that the program was of benefit to both populations. However, there were shared concerns regarding the transport, safety, and outcomes of the program for participants. Based on our findings we offer several policy considerations in the implementation of intergenerational programs.

## 1. Introduction

Embedded in history is a norm that families should look after their own [1,2,3]. To a large extent, this work has been undertaken by female family members [4]. More recently, increased financial pressures and changes in the workforce participation of females has resulted in more women accessing social services for the care of their children and elderly [4,5,6,7]. The increased demand for care services, combined with the ageing population has created a ‘sandwiched’ generation in which families are increasingly managing care for both their elderly parents and young children at the same time [7]. Inevitably, these changing patterns in care have put significant strain on the social sector [6,8,9,10]. One care option that has been generating great attention, is the provision of intergenerational practice programs.

Intergenerational practice programs are programs designed to meaningfully connect two or more generations. These programs allow organizations to facilitate a shared interactive program that is designed to provide meaningful social roles to participants, and to transmit knowledge and skills between generations [11]. In doing so, intergenerational practice programs provide purposeful interactions between older and younger cohorts over a defined period, in a shared space. Yet, while research on these programs has demonstrated improved social and behavioral outcomes for both older people and young children [12,13], no research has examined public perceptions of these programs. As such, this paper, through the voices of experts and consumers of aged care and childcare, reports on public perceptions, highlighting the potential barriers to the uptake and implementation of intergenerational practice programs. In doing so, this paper makes an important contribution by identifying key attributes that must be addressed when developing an intergenerational practice program. The identification of these attributes is important in the context of demonstrated demand for innovative programs to fulfil the needs of both older and younger populations in Australia.

### Intergenerational Policies

Aged care and childcare policies in Australia have largely been segregated due to expectations regarding care outcomes for the young and old. For example, aged care focuses on providing quality care and support to the elderly, whereas childcare focuses on providing safe and high-quality educational outcomes for children [14,15,16,17]. More recently, there have been calls for a revisitation of these expectations and development of renewed policy agendas that allow for the introduction of intergenerational practice programs [16,18,19].

Intergenerational practice programs have been found to improve children’s attitudes towards dementia, knowledge of dementia, and engagement in activities [20,21,22]. However, despite this, intergenerational practice programs that have a sole aim of children interacting with older adults with dementia or related cognitive decline illness, are limited [20,21]. Older participants reported having an enjoyable experience and recounted that they were more engaged in other activities beyond the program because of their participation [20,21].

Intergenerational practice programs have also been found to increase morale and motivational needs of participants, and to increase happiness and mobility for older people living in residential care. For children, intergenerational practice programs have resulted in improved confidence and increased awareness of older people around them [22,23,24]. In addition, participating children have been found to have higher personal and social development scores than those who have not participated in the program [22,23,24].

The introduction of formal childcare programs has improved the reading rates of children accessing these services, empowered working parents to stay working and, therefore, to be productive members of society [22,23]. In addition, research has found that formative experiences of children play a significant role in the outcomes they experience in life [25,26]. Consequently, it is likely that participating in intergenerational practice improves the social outcomes of children in the long term [25,26]. Yet, while formal reviews of intergenerational programs find that participation in any care program (such as aged care or childcare) is often inhibited by the demand, attractiveness, and cost of care services [5,27,28,29], there has been limited empirical research conducted to explore additional social care program choices, such as intergenerational programs at a policy level. Further, even fewer studies explore the potential pushbacks these programs may face if implemented. Therefore, this study is important to address these concerns and maximize the uptake of the program, as well as to encourage the co-design of services in society [30]. This study addresses this gap in the literature by being the first to explore the reasons for, as well as against, uptake of intergenerational practice programs in Australia and was designed to answer the research question: What are the barriers and supports to creating an intergenerational practice program in Australia?

## 2. Materials and Methods

To investigate the key attributes that need to be considered when developing an intergenerational practice program in Australia, data was collected from two populations. First, a Delphi process was conducted with 23 experts in aged care, childcare, social policy, and delivery of services in these areas. Stakeholders interviewed included the following: academic experts in the fields of aged care, dementia care, human resource management, childcare and strategy/organizational behavior (n = 5, 22%), aged care practitioners with at least 5 years’ experience in the field as managers or clinicians (n = 11, 48%), child care practitioners with at least 5 years’ experience as educational leaders or managers (n = 5, 22%), and consumers (n = 2, 9%). These experts were identified due to their prestige and reputation in the field.

Experts were invited to participate in focus groups between November 2018–July 2019 and the research team sought to ensure that each focus group had representatives from each area of expertise to facilitate a dynamic exchange of ideas. Each focus group took approximately 90 min and was typically conducted in person with three experts. Individual discussions did take place where the expert was unable to participate in the focus group. No compensation was provided for the time of experts as they volunteered for the study. Following this, a national survey was distributed to the general population to explore the feasibility and demand for intergenerational practice in Australia. For ease of readability, this paper presents the methods of data collection separately; however, data sets were blended during analysis.

### 2.1. Delphi Process

The Delphi process was structured around iterations. In the first iteration, participants were provided with a set of readings for perusal prior to a panel discussion of the possible advantages and disadvantages of these models. The readings provided were drawn from literature [31] related to various intergenerational practice models that already existed related to the provision of care services delivered in a community setting.

The literature identified four models of care that could be implemented based on a review of the legislation in Australia: visitation, co-location (visitation or shared space), and single site. Visitation models provided childcare and aged care services on separate sites (and potentially by different organizations), with the younger population typically leaving their place of residence to visit the elderly. Co-located models offered intergenerational practice programs in locations that were already sharing space in the delivery of aged care or childcare services, either on the same ground or site. Co-located visiting models included provision of services on the same site but where there was no shared ground and, as such, one or both groups must walk to a shared area to participate in the program. Co-located shared site included those that already shared their spaces as part of the facility. Single site programs included those that provided both aged care and childcare models together as part of one program.

To conduct these panel discussions, two interviewers were present and used open-ended questions to gather the panelists’ opinions on the attributes of each of these programs. The panelists were asked to consider the relevant policy that existed and the meaning these attributes were proposed to have to participants. The legislative boundaries were identified by previous research [16], which was provided to the panelists as part of the pre-reading material. Panelists were asked to comment on the following questions:From the perspective of a consumer/agency/government that you represent, please rank the models in order of preference, listing your most preferred model on top.How do you feel about each existing model?What needs to be overcome to deliver each model effectively in the Australian setting? Think about the attributes of each model—for example workforce issues, such as skill set, and award issues.To deliver intergenerational practice programs, comment on the accreditation standards that must be adhered to.In addition to the current models, can you think of a hypothetical intergenerational practice program model that would be suitable in Australia? Please explain as comprehensively as you can your ideal model and the requirements of your model.What issues need to be overcome to achieve your hypothetical intergenerational practice program model for the Australian setting?Listing other attributes that you believe must be considered in the development of each model.

Attributes that were pre-identified as areas for discussion from the literature included the following: age of participants, curriculum, activities, occupational therapy, workforce (skill set, award issues), staff ratio, accreditation standards, and opening hours. Participants then added attributes as they saw fit as part of the discussions.

The second iteration saw the views gathered in the first iteration presented back to the wider group for further comment and consultation. After which, this feedback was gathered and re-presented to the focus group panelists in the third iteration, which focused on the practicalities and client needs that may need to be considered. This process led to the identification of two models that were feasible and practical in an Australian context: The Shared Campus model and the Visiting model. These two models were chosen by experts as feasible models based on their legislative ability, economic feasibility, safety, benefits, logistics, and community acceptance of the intervention in these areas. These models and attributes were then used, in addition to a literature review of peer-reviewed articles, and additional qualitative research, to create a survey that was then distributed to the wider Australian public.

Panelists were approached via email because of their expertise in their area of interest, and all participants who were approached responded positively to the invitation.

### 2.2. National Feasibility Survey

The survey included questions that asked participants to assign a value to each of the two intergenerational practice models presented. The survey asked questions about participants’ willingness to pay and to explore the demand for intergenerational practice (See 35 for more details on this survey creation). To examine the key attributes required to be considered before creating an intergenerational practice service, participants were asked the following open-ended question *“If your current care service included an Inter-generational Care program, with either a Shared campus or Visiting campus model, for the SAME price as your current care, would you use this service? If not, why not?”.*

#### 2.2.1. Procedure

The survey was widely distributed via in-person (4%), Facebook (1%), and online * options (95%). Snowball sampling was used to maximize the reach of possible participants through the experts engaged in the Delphi process, as well as industry partners. Surveys were also distributed online, supported by a variety of stakeholders using the program Qualtrix to administer the survey. A total of 1445 surveys were returned to the researchers. Of which, 1020 participants responded to the question posed. The eligibility to participate in this survey included: (a) be able to understand English, and (b) have a child in childcare or an adult accessing aged care services at the time of the survey. These criteria were chosen as we were interested in the views of current aged care and child care participants to provide insights into whether intergenerational practice would be of value to them at the time of the survey.

#### 2.2.2. Analysis

The responses to open-ended questions within this survey were analyzed using a process of reflexive Thematic Analysis (reflective TA) [32,33], whereby themes are constructed through a subjective sensemaking of recurrent patterns in the open coding process. Throughout the open coding process, a semantic approach was favored with initial codes mirroring the language of participants. These semantic codes were then reviewed to identify latent codes that better captured the underlying meaning. As an example, the response: *“I think this would be wonderful and would benefit both generations… as long as the safety of the children is at center stage and children are not left alone with “non-carers”. That said, I would welcome this”*, was initially semantically coded as, *“It would be wonderful but safety is an issue”*. Following a process of peer-validation [34,35], in which the research team discussed the coding framework, this passage was latent coded as: ‘Safety’ and ‘Supportive’. Themes were then created by identifying consistent patterns as evidence of shared concerns. Intercoder reliability was not measured, as this was deemed inconsistent with the reflexive approach taken [35,36]. However, the use of the peer-validation meeting and iterative discussion with the Delphi group supported the authenticity of the themes presented here.

#### 2.2.3. Ethics

Ethics approval was received from the university (Griffith University Human Research Ethics Committee (GUH- REC Approval Number AFE/21/13/HREC). Written consent to take part in the Delphi process was obtained and returning the questionnaire to the respondents was considered consent for the survey.

## 3. Results

The results of this study revealed that 71.3% of respondents (N = 1020) would use intergenerational practice programs if they were available. However, there were concerns raised by both experts and potential consumers in creating an intergenerational practice as formal policy in Australia. This section presents these concerns by initially analyzing the shared concerns (i.e., brought up by both experts and potential consumers), before identifying the unique concerns raised by the consumers and experts in this study.

### 3.1. Shared Concerns

There were four key shared concerns raised by all participants in this study. These were: 1. Logistical concerns around transportation. 2. Occupational Health and Safety (OHS) considerations such as infection control, behaviour management of older participants. 3. Ensuring the safety of all participants; consideration of the activities that were being implemented; and, 4. Considering personal factors that may impact on the desired participation in the program, such as personality and preferences.

#### 3.1.1. Transportation

All participants felt that transporting the elderly to the young would be the most practically feasible way of bringing participants together in the visiting model. Specific concerns were raised about participants’ ability to walk, as well as the practical consideration of car seats, raised by both experts and potential consumers. Specifically, one childcare expert reported:

*“(To) fit in with what the needs are and logistically for us to get a bus and have 20 car seats and all the requirements…for us it suits us better to have the (elderly) coming to our immediate complex”* (Child Care Expert)

In addition, an aged care expert added that transporting the elderly to the young would also improve social relationships between residents and their community:

*“Most of the facilities have the bus and for the reasons, we can get our residents on the bus and see some with mobility, and our residents are required to maintain the friendships and their community involvement”* (Aged care expert)

However, programs with people with dementia may need to further consider the logistics of this with one participant noting that:

*“It really depends on the group. Because for some older people they’re really excited about being in a new environment. But if they had dementia for example, taking them to a facility that was developed for young children. It might take them a while to be orientated and there might be some other issues with it. So, it would be dependent on the group of older people I think”* (Government policy expert)

In addition, one parent was concerned with the safety and logistics of transporting older people as she reported *“Taking frail aged people out of their home can be difficult.”*

Therefore, further consideration regarding the feasibility of who travels to whom is needed in visiting models of intergenerational practice, with particular attention and risk assessments completed considering the participants involved. In addition to transport logistics, all participants reported concerns regarding occupational health and safety considerations.

#### 3.1.2. Occupational Health and Safety Considerations

Concern around infection control and the safety of participants were raised by all parties. In particular, concerns around infection control were rife across all participants, with the one expert commenting:

*“There’s something about when they’re little, obviously, they are themselves getting exposed to different viruses. You don’t even know when you got measles and then suddenly an old person is just about to get shingles, so there would be some infection control things also in there that would be particular… Scarlet fever, there’re a few biggies that are now around those little-ies.”* (Social worker)

In addition, two potential consumers reported concern around their own health and the health of their child, regardless of their desire for the program:

*“My other reason for saying no is that my immune system is compromised and being in a child care environment would not be a safe place for me.”* (Aged care recipient)

*“Older people are bigger and they been living for 65 years they might have contact with a lot of people and can have more disease that is why I’m a bit hesitant to change his usual from child care to preschool setting now to this inter-generational care program.”* (Parent)

These findings identify that infection control policies need to be a priority when implementing this policy, with both children and staff required to follow these strict processes in order to protect both parties.

In terms of safety, behavior and the potential for sexual/physical abuse of participants was raised by potential consumers, as well as experts. For example, an expert on dementia care commented:

*“What concerns me the most is that, particularly, with people with dementia, they have behavioral problems that could create issues, so you need to think about how that’s going to be managed.”* (Dementia expert)

In addition, a couple of parents reported concerns about the exposure to possible sexual abuse by older participants, and their desire for closely supervised programs. Specifically, one parent reported:

*“My only worry would be if my child was abused by an older person, you said they’re screened and it would be supervised but this sort of thing does happen and it would be a concern of mine.”* (Parent).

Interestingly, an older person also reported that they were concerned about this specifically, by stating,

*“(it’s) too risky these days of being accused by some parents that you may be interfering with their child and it only has to be the slightest rumor and you are presumed guilty. As a result, they would not participate in the program.”* (Aged Care Recipient, male).

While another parent reported that they would be happy to join in the program, *“with appropriate risk/compliance measures in place”*.

These findings highlight the importance of screening and blue cards for participants in signaling the safety of the program to potential participants. Due to the nature of progressive cognitive decline, regular screening is needed of participants which would ensure constant monitoring existed. Where such screening may not be possible, the implementation of a closely supervised program would be needed to alleviate safety concerns. Another important safety signal would be the presentation of an exit plan to support participants that no longer meet entry requirements

#### 3.1.3. Specific Activity Plans

In addition to the transport and OHS issues identified, all participants reported that a specific focus was needed on the types of activities that underpin an intergenerational program. For example, a dementia and ageing expert reported that there was a need to individually tailor programs to the participants that are recruited because:

*“Somebody with younger onset that is quite high functioning. Wants to participate in what they see us something that’s purposeful and meaningful and something they enjoy. They want to choose whatever that is, and it might be that they want to go and do so some gardening, or they might want to go out to an art gallery. So, they may not want to have visits, but their activities might be quite different from other people in the group.”* (Dementia and ageing specialist).

Similarly, a social worker commented that it is not necessarily about the age of participants but how participant programs are tailored to address the strengths of the group:

*“It’s about people’s needs and the strengths and how they’re supported. Age is irrelevant enough in a way, it’s around what individuals need and how you make that individual group setting. I think that’s our greatest challenge out there.”* (Social worker)

However, a parent reported their hesitation in joining the program due to the perception that the activities planned may detract from the educational outcomes of children. Specifically, one parent reported that:

*“I seek specific early childhood education for my child’s development. I feel integrated care detracts from each group’s needs.”* (Parent)

These findings suggest that spending time focused on planning a curriculum that is tailored around educational and care outcomes for all participants is critical in achieving meaningful engagement in the program.

#### 3.1.4. Personality Variables

Finally, there was an acknowledgment that this program was just not for everyone, reported by all participants. Specifically, an aged care expert reported:

*“All the residents that I take to the child care want to be there, but not all elderly people like children and not all children like the elderly. So, you’ve got to know the clients and know that they are willing participants. The children are willing participants. Because you can’t just be all warm and fuzzy and say; old people love children and all children… Because they don’t. It’s just the way of growing up.”* (Aged Care expert).

These feelings were also reciprocated by respondents in the survey, where one older person admitted that:

*“I no longer have much patience with young children.”* (Older person)

Whereas, another older person reported,

*“I get too tired to cope with any interaction between myself and young children for 1–2 h a day.”* (Older person)


*Similarly, one carer reported that this program was not beneficial for her significant other because “he hates children”.*


From these findings it is clear that, combined, there are some barriers that need to be overcome and addressed for both parties prior to the introduction of this program. However, in addition to the barriers that were identified collectively in this study, the experts also had some additional concerns that need to be considered.

### 3.2. Concerns Raised by Experts but Not Potential Consumers

Some additional logistical concerns were raised by experts in the form of staffing, as well as the marketing and timing of the program.

#### 3.2.1. Staffing

Staffing concerns were raised about the training competencies of staff to cope with both an older and younger participant pool at the same time. Specifically, experts raised the idea about the need for specific competencies in this area:

*“Young children require skills in terms of human development over time, in terms of, teaching, learning, and looking at different activities. Whereas, older people, particularly those with cognitive decline, staff will often treat them as a child when that’s… They’ve got capabilities, capacity that young children do not have. I think unless you understand the aging process, you will treat the older person like someone who can’t do anything.”* (Dementia expert).

In addition, another dementia expert noted that:

*“There is a still a lot of ageism and even though people might say, “I love older people.” Young babies, young children will take priority as society tends to focus on younger people rather than older people. My concerns are that if a baby is crying or a young child cries something, that’s going to take priority over an older person who may be sitting in diapers, for example.”* (Dementia expert)

This suggests that there is a need for a coordinator role that does have the multiple scopes of practice, or a team that can work together to create a meaningful program that addresses any underlying ageing biases. This was further highlighted by one workforce expert who noted that:

*“You want to resource some sort of a coordinator that can go, “Okay, I’ve got an aged care here and I’ve got a childcare here, you’re in the same area, they’ve got a bus, you do not have a bus, so I can hook you up.” And the childcare and aged care start there, rather than have either an aged care facility or a childcare facility suiting around ringing 30 different other facilities in their region to find one that might be interested in getting on board.”* (Workforce expert).

Consequently, finding a team that can work together cohesively to create a purposeful program, and creating a vocational education training certificate IV level competency program on intergenerational practice is needed to address some of these training and support concerns.

In addition to staffing competencies, training around death and dying is needed. However, not only is training required to support parents to talk to their children about death if a participant dies, but some processes are needed organizationally to train staff in this competency:

*“One of the other things and this is probably one of the greatest concerns and I don’t know how, but when you have children become fond of particular older people or vice versa, they become fond of the younger children and either the older adult dies or the child goes. How do you manage that? I think that children are pretty adaptable to death at a young age, but I think it’s still needs an element of what is actually happening and why they have gone. The parents would need to understand that this is going on as well.”* (Dementia Expert)

#### 3.2.2. Marketing and Timing of the Program

Finally, the marketing of the program needs to not call it a “day care” program, as one dementia and aged care expert reported that, “People with dementia quite often get irritated by someone saying they are going to day care.” This suggests a marketing solutions panel may be required to pilot the name of the program if the government was to support the formal introduction of intergenerational practice into the social care policy and programs.

In addition, careful thought is needed regarding the timing of the program. Specifically, to fit into the “sleep cycles and morning tea” of participants. This has to consider travel time in either of the models to ensure the normal daily activities of participants are not impacted significantly by this program.

## 4. Discussion

The provision of care as a formal program in Australia has long been considered a social ‘good’ [37]. However, childcare and aged care policies are often segregated, despite the increase in a sandwiched generation who are faced with the pressures of looking after their elderly parents and young children. In addition, childcare and aged care are in high demand, yet often do not always meet the needs of the end user [6,27]. In addition, the cost of care for both the elderly and children prevents some populations accessing these services [5,6,28].

Globally, we have seen a shift from the traditional model of intergenerational families to more dispersed families, because of increased financial pressures and migration [38]. Consequently, an increased focus on alternative models of care, such as intergenerational practice programs, which are designed to support families looking for alternative models of care to meet their needs, has emerged.

While intergenerational practice programs have been gathering much interest in recent times from the media and public in Australia, as well as in the empirical literature, this study was the first of its kind to explore experts’ and potential consumers’ perceptions of these programs, specifically examining whether individuals would use intergenerational practice programs and why. In doing so, this research found a large percentage of participants would use the program if it was offered as a formal offering. For those who would not use the program some clear reasons emerged.

This study demonstrates that consideration is needed about the transport of participants to each venue, and, while practically it would be easier to have the elderly visit the young, this may depend on the needs of participants. Therefore, this should be an individual program level decision made earlier on in the development. In addition, consideration and preparation of clear infection control guidelines are needed to protect the physical health of all participants, as well as the careful screening of participants with consideration to gathering blue-cards for all participants prior to participating. The concern around risks was also found recently in a study involving stakeholders interested in connecting schools to retirement aged homes [39] Moreover, a carefully considered education program, communicated to all parties prior to entering the program, is needed, as is an acknowledgement and respect for the fact that, simply, this program is not for everyone.

From an organizational perspective, consideration to the development of grief and loss training for staff, as well as the development of a Certificate IV level qualification in intergenerational practice, to breakdown ageing perceptions and ensure protection of the rights of all participants are needed. Finally, the marketing and timing of the program would need to be considered at organizational and program levels to ensure the best outcomes for all.

What is interesting, is that while this study highlighted issues for consideration when developing policy, we know that the successful provision of intergenerational practice programs relies heavily on emotional and social support [40,41,42] Yet, access to these services remains limited to those who can afford services [42].

Indeed, “if ‘good care’ depends disproportionately on the quality of the care relationship” [37] (p. 1), then, more attention needs to be given to the quality of relationships developed in care programs, as well as the workforce, a sentiment shared in the current literature on intergenerational programs in Australia [39]. For the purposes of this paper, though, the provision of ‘good care’ should result in an increased focus on intergenerational practice relationships, which has been found to improve the social and behavioral outcomes for all participants. In doing so, educational outcomes can follow.

## 5. Conclusions

In conclusion, while further research is needed to explore the economic viability of intergenerational practice programs, this study has presented some key findings that allow policymakers to consider the feedback from both experts and potential consumers of this service in the future. This study identified several policy implications to be addressed. Building on previous research on policy implementation in the UK [41], to implement intergenerational practice in Australia consideration is needed about the organization and oversight of these programs to ensure safe outcomes for all participants, as well as the voluntary inclusion of participants in these programs, as it is not suitable for all. In addition, like previous research, this study found that while managers play a key role in selecting appropriate residents and closely monitoring the outcomes of these activities, so too do residents and children by continuing to recognize what works and does not work when formulating activity plans for these interactions. Consequently, programs that address these concerns, as well as the concerns about availability, affordability, and quality raised by researchers [42], are likely to succeed.

What this study has not identified yet is the consideration to cost. In line with previous research [5,28], while intergenerational practice programs are one alternative to the existing programs that can meet the demands faced in care arrangements, if it becomes unaffordable, it will not meet the need. While previous studies [28] have begun to explore these cost considerations in their demand estimation of intergenerational practice, further research is needed to explore the actual costings and the pricing point that organizations can use to ensure the ongoing sustainability of these programs in the future.

## Data Availability

Not applicable.

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
