# Peer review of "Key Considerations to the Introduction of Intergenerational Practice to Australian Policy"

_ijerph, 2022, doi:10.3390/ijerph191811254_

Round 1

Reviewer 1 Report

Review for “Reviewing the Australian government policy considerations needed for the implementation of a formal social intergenerational practice program”

1)    Thank you for the opportunity to review this paper. This study explored expert and consumer perceptions of intergenerational programs. A better understanding of perceptions regarding intergenerational contact are essential for creating conditions to facilitate and promote intergenerational contact.  

2)    The title of the manuscript is quite long – perhaps it could be simplified? For example,  “Australian government policy considerations for a formal social intergenerational care program”

3)    A definition of “intergenerational practice programs” (pg 2 – line 63) would be useful. Are “intergenerational practice programs” the same as “intergenerational care programs”? If so, please include this information. If not, please provide information that distinguishes the difference(s) between the two terms. 

4)    Page 3 (line 134) – “To deliver IGC, comment on the accreditation standards that must be adhered.” Please indicate what IGC stands for when first using the acronym.

5)    Can more be said about why the shared campus and visiting model were identified as the model feasible models? Please elaborate on why these two models were deemed feasible. 

“This process led to the identification of two models that were feasible and practical in an Australian context: The Shared Campus model and the Visiting model” (pg 4).

6)    In the procedure section on Page 4, more information about the stakeholders whom completed the survey would be useful. Perhaps a breakdown of positions, industries, etc. that said stakeholders work in or how they are connected to intergenerational care programs could be included.

In addition, more information on the potential consumers would be useful as well. Who were these participants? How was their potential consumer status determined? What sort of inclusion/exclusion criteria was used for these participants? 

7)    Were participants compensated for their time?

8)    Was interrater reliability calculated between the two researchers who conducted the coding (Analysis – page 4)? Please include this information. In addition, how were the coding categories determined? More information about the coding would be helpful. 

9)    When was the data collected? Please include this information in the methods section. For example, was the data collected during the COVID-19 pandemic? Given the concerns about infection control (pg 6), I’m wondering if specifics around COVID-19 infection control came up. If the data was collected before the COVID-19 pandemic, it would be helpful to note. 

10) Most participants reported interest in an intergenerational care program if available. Is there any existing research on the potential discrepancy between behavioral intentions (specific to utilizing such social programs) and actual behavior? This may be another area of future research, but I am curious if behavioral intentions are a good predictor of using these types of services or programs when available. 

11) In summary, this research examines an important question about the possible strengths and weaknesses of intergenerational care programs in Australia. Further research is needed to determine the feasibility of such programs (e.g., cost). 

I applaud the authors for investigating this important issue and wish them the best of luck in the future.  

Author Response

Thank you for your comments. Please see attached document for our revisions.

Reviewer 2 Report

Overall, this is an interesting, well-written and timely paper exploring what consumers and experts think about intergenerational care. A recent paper in the Australasian Journal on Ageing documented similar benefits and risks in the Australian context (the co-location model only though), and so might be worth reference – reference below. One question about the method: the Delphi Process is interesting, I just wanted a few more details about how this was conducted – online or in-person, meeting length, and time between meetings. The results are interesting, and should inform the future design and communication of intergenerational care programs. Slightly more details re the why and how of thematic analysis – including a citation of which authors inspired this paper authors (eg is it reflexive thematic analysis) – should be included. 

RECOMMENDED REFERENCE: Trotter, MSanders, PLindquist, MMiller, EHajirasouli, AsoBlake, A, et al. (2022). Intergenerational living and learning: The value and risks of co-locating retirement villages on secondary school campuses – Evaluating the GrandSchools visionAustralasian Journal on Ageing, online

Author Response

Thank you for your comments. Please see attached our response to your helpful suggestions

Round 2

Reviewer 1 Report

Thank you to the authors for providing addition information throughout the manuscript. I do have two remaining questions from my original review that were not addressed in the first round of revisions. I have included them below:

5)    Can more be said about why the shared campus and visiting model were identified as the model feasible models? Please elaborate on why these two models were deemed feasible. 

“This process led to the identification of two models that were feasible and practical in an Australian context: The Shared Campus model and the Visiting model” (pg 4).

6)    In the procedure section on Page 4, more information about the stakeholders whom completed the survey would be useful. Perhaps a breakdown of positions, industries, etc. that said stakeholders work in or how they are connected to intergenerational care programs could be included.

In addition, more information on the potential consumers would be useful as well. Who were these participants? How was their potential consumer status determined? What sort of inclusion/exclusion criteria was used for these participants? 

Thank you again for your careful review and edits!

Author Response

Please see attached file for responses. Thank you for allowing us to clarify our research further. 
